# Viscoelastic Analysis of Pectin Hydrogels Regenerated from Citrus Pomelo Waste by Gelling Effects of Calcium Ion Crosslinking at Different pHs

**DOI:** 10.3390/gels8120814

**Published:** 2022-12-10

**Authors:** Tu Minh Tran Vo, Takaomi Kobayashi, Pranut Potiyaraj

**Affiliations:** 1Department of Energy and Environmental Science, Nagaoka University of Technology, 1603-1 Kamitomioka, Nagaoka 940-2188, Niigata, Japan; 2Department of Materials Science, Faculty of Science, Chulalongkorn University, Bangkok 10330, Thailand; 3Department of Science of Technology Innovation, Nagaoka University of Technology, 1603-1 Kamitomioka, Nagaoka 940-2188, Niigata, Japan

**Keywords:** pectin hydrogel, viscoelasticity, pomelo albedo, Ca^2+^ crosslinking, pH-effect, gelation

## Abstract

Pectin was extracted from citrus pomelo waste, and the effects of calcium ions (Ca^2+^) on the gelation and hydrogels properties were investigated over a pH range of 3.2–8 by using viscoelastic analysis. The gelatinization of Ca^2+^-pectin was examined at concentrations of 0.9, 1.8, 2.4, and 3.6 M of Ca^2+^ in aqueous pectin solutions of 1%, 2%, 3%, and 4%. The gel transition of Ca^2+^-pectin solution to hydrogels was determined by measuring the storage modulus (G’) and loss modulus (G”) under mechanical strain from 0.01 to 100%. In a hydrogel of 3% pectin at Ca^2+^ = 2.4 M, as pH increased to 7, the G’ at 0.01 strain % was 3 × 10^4^ Pa, and 3 × 10^3^ Pa at pH 5, indicating that the crosslinking weakened at acidic pH. Due to the crosslinking between the calcium ions and the ionized carboxylic acid groups of pectin, the resulting hydrogel became stiff. When the mechanical strain % was in the range of 0.01–1%, G’ was unchanged and G” was an order of magnitude smaller than G’, indicating that the mechanical stress was relieved by the gel. In the range of 1–100%, the gel deformation progressed and both the moduli values were dropped. Collapse from the gel state to the solution state occurred at 1–10 strain %, but the softer hydrogels with G’ of 10^3^ Pa had a larger strain % than the stiffer hydrogels with G’ of 10^4^ Pa.

## 1. Introduction

From the perspective of the biorefinery, the use of biomass polymers is of interest as an alternative to petroleum-sourced polymers, and the recycling and effective use of food waste have become necessities due to a growing need to build a sustainable society [1]. Increasing demand for citrus consumption has been growing and fruit waste is a relative consequence which is causing an environmental burden. Conversely, citrus by-products are well known as promising bio-resource materials [2]. Recently, the utilization of citrus waste has served not only in bioactive compounds but also in biofuels [3,4]. The utilization of these bioactive ingredients as a material is expected to lead to new developments. Among them, pectin is a polysaccharide which is extensively dispersed in plant cell walls [5], is one of the essential components, and is a macromolecular anionic polymer having a backbone of (1,4)-linked-D-galacturonic acid units with methoxyl groups [6]. Due to the functional qualities that pectin possesses, it has been extensively studied for food additives, personal care, and pharmaceuticals and is widely used industrially, for example, in thickening, gelling, and emulsifying [7]. In these applications, pectin hydrogels have been used as matrices of drug release and drug delivery [8]. Because the galacturonic acid segments in pectin hydrolyze to negatively charged -COOH groups above pH 3.5, crosslinking ionically mediated via divalent cations, such as Ca^2+^, forms “egg-box” crosslinks, which facilitates the gelation of pectin [9,10,11]. In recent research, Ca^2+^ ions engaged monocomplexes with single, dissociated carboxyl groups, leading to random crosslinking between the pectin chains [12]. Unlike other natural polysaccharides such as cellulose and chitin, pectin is a water-soluble polymer because pectin contains ionized carboxyl groups due to the hydrolysis of the methoxyl group of the galacturonic acid group. Therefore, there are myriad problems which can be overcome by mixing divalent ions as a bridge connecting these pectin chains to form a gel network [13,14]. Below pH 3.5, acid-induced gelation was observed, but Ca^2+^-induced gelation was efficient for fully charged chains at pH > 4.5 [15]. As intrinsic factors for pectin gelation, the degree of polymerization [16] and the percentage of methoxylation in the galacturonic acid segments [17] influenced the gel behavior. There were some cases of pectin–calcium mixtures exhibited for gelation. For example, the stiffness of pectin hydrogel was affected by the degree of methylation, as indicated by the storage modulus value depending on the mechanical deformation rate (strain%) [18,19]. However, until today, no specific results were given for the elastic behavior of pectin hydrogels. Generally, there has been a way to accelerate the gelatinization of pectin, which is achieved by combining a polymer solution with divalent ions followed by the gelling occurring throughout the whole volume of the aqueous mixture [20]. Previous research also examined the effect of pH from 7.0 to 3.0 on LM pectin from okra pods. The pH change was affected in the galacturonic acid polyelectrolyte conformation rearranging from extended form at neutral pH to compact conformations at acidic pH. This caused early vitrification at neutral pH when galacturonic acid residues dissociated [21]. Furthermore, pectin showed a delay in structure formation, as the pH varied to acidic region. At lowering the pH decreased the number of possible calcium ion binding sites, inhibiting gelation in the presence of sugar [22]. The amount of Ca^2+^ or pectin concentration also influenced the gel strength in sugar-containing pectin gels, indicating that a higher Ca^2+^ content had a higher modulus of elasticity and a greater degree of crosslinking [23]. Therefore, the study of pectin gelation and its hydrogel was mainly indicated by its development into applications. Thus, there is still little fundamental study adapting viscoelasticity analysis for the gel state. Although the swelling properties of pectin hydrogels in water [24] are known, the details of viscoelasticity have not yet been reported. Additionally, aqueous solution pH seems to affect pectin gelation due to the presence of COOH groups, showing COOH ↔ COO^−^ + H^+^. However, little was known about the influence of varying pH values in calcium-pectin solution and resultant hydrogel properties. In addition, the gel’s properties are essential for the elucidation of gelling behavior using viscoelastic analyses.

Therefore, to clarify what happens to the gel state when these gelation factors change, the goal of this study is to look into the effect of pHs from acidic to alkaline conditions at different calcium ion concentrations for the gelation process and the resultant pectin gels using viscoelastic measurements. Here, the pectin sample was extracted from pomelo citrus waste and utilized for the fabrication of pectin hydrogels.

## 2. Results and Discussion

### 2.1. Gelation Behavior of Aqueous Pectin at Different pHs

After extraction of pectin from pomelo citrus waste, the solid polymer of a yellowish coloring was examined by FT-IR analysis. The spectra indicated 74% methylation of the resultant pectin. Then the lyophilized polymer was dissolved in aqueous water. The titration of 20 mL of 1% (*w*/*w*) soluble pectin solution by NaOH solution indicated that the dissociation of the proton of COOH to COO^−^ occurred from pH 4 to pH 9 (Figure 1a). Thus, the carboxyl groups of pectin can be either dissociated or non-dissociated depending on pH, which affects the polymer charge. In addition, the values of the zeta potential were measured at different CaCl_2_ concentrations, as shown in Figure 1b. Apparently, the COOH groups dissociated and provided a negative charge in the solution. The effect of pH and CaCl_2_ addition on the zeta potential of pectin solutions presented negative values. As the pH increased from 3.2 to 4.8, the value was tented to shift toward the negative region and then remained pretty steady in the pH 4.8–8.9 range. Especially in the range of pH 3 and pH 5, the minus value of about −15 mV changed to −30 mV when the pH was from 3.2 to 4.8, respectively, in the case of the absent calcium ion. Related to the neat pectin solution, as the calcium concentrations were varied from 0.9 M to 3.6 M, the zeta potential values were shifted toward negative region from 2.5 to −13.3 mV at pH 3.2. Additionally, at pH 8.9, those were observed at −5 mV and −30.6 mV for [Ca^2+^] = 3.6 M and 0 M, respectively. Remarkably, the highest Ca^2+^ concentration at 3.6 M exhibited to fall to zero charge at pH 4, indicating salt shielding effect [25,26] of the added salt to the COO^−^ group. These evidenced that the electrostatic forces were neutralized with the addition of Ca^2+^ ions. In addition, zeta potential values tended to decrease the value with the increasing of the concentration of calcium ions. This meant that the electrostatic shielding effect worked efficiently on the ionized pectin with increasing calcium ion concentrations.

For pectin solutions having different concentrations of Ca^2+^, the gelation of pectin was studied using dynamic viscoelasticity. Figure 2a displays the time change of storage and loss moduli in the range of pH 3.2 to 8 at 2.4 M of Ca^2+^ concentrations. Here, the pectin solution was fixed at 3% in each aqueous solution and the strain % for mechanical oscillation was 1% at 1 Hz. The time dependence of the G’ and G” moduli increased from their started value at time zero, indicating that efficient gelation occurred at pH 7. As seen in Figure 2b, the value of tan δ dropped sharply at a certain time and then became constant. At the point of time when tan δ = 1, the aqueous pectin solution solidified from liquefaction. Thus, the gel time was defined as the time at tan δ = 1. As evidenced from the data, less gelation occurred at pH 3.2 during the observed time. When the pH increased from pH 3.2 towards pH 8, the trend of increasing G’ values became higher. Namely, the gelation of pectin–Ca^2+^ solution occurred when the solution was shifted toward basic pH. In the gelling process, the G’ value significantly improved from 299 Pa to 737 Pa within 55 min at pH 5 and 7, but decreasing pH to 3.2, G’ was about 1 × 10^2^ Pa, meaning the gelling ability was much lower. In acidic conditions, the titration indicated less dissociation of COOH groups on the pectin backbone, limiting crosslinking ionically by cationic calcium ions for the gelation. This might be the reason for less gelatinization due to lower dissociation to COO^−^ of the galacturonic acid group in the pectin. As presented in Figure 2a,b, the large G’ values suggested that gelation robustly formed in the pH 7–8 range. In the pectin gelation, Ca^2+^ concentration is another considerable parameter. Figure 2c,d depict time change of G’, G”, and tan δ in different Ca^2+^ concentrations at pH 7. The gelation was progressed in Ca^2+^ = 2.4 M and 3.6 M, because the G’ values were enhanced in the cases of higher calcium concentrations. However, the gelation times estimated from Figure 2d were 4.3 and 5.6 min, increased from 2.4 M and 3.6 M, respectively. The quantitative relationship between ionized COO^−^ and calcium was almost the same at Ca^2+^ = 2.4 M. Accordingly, the gel intensification might become high at the equivalent condition. 

### 2.2. Viscoelastic Properties of Pectin Hydrogels

As seen in Figure 3, the appearance pictures of calcium–pectinate hydrogels looked like they highly retained water in the gelling matrix. As listed in Table 1, the values of water contents were in the range of about 1000–2000% in the dry base. When the pH of the pectin solution increased from pH 5 to 8, the values of water content varied from 1558% to 1207%, respectively, resulting in a tight gel formed at higher pH. In addition, when calcium concentrations increased at pH 7, the water content also decreased. Furthermore, the water content showed a decreasing trend at pH 7 with an increasing of the pectin concentration.

For these pectin hydrogels, the viscoelastic behavior was measured at different mechanical strain % in the range of 0.01% to 100%. At each oscillatory deformation to the pectin hydrogels, the storage modulus (G′) and loss modulus (G′′) obtained were plotted against the strain % (Figure 4). When pH was changed in the range of pH 5 to 8 for 3% pectin solution with 2.4 M Ca^2+^ concentration, the formed hydrogels had G’ and G”, as seen in Figure 4a. There were constant values observed at strains ranging from 0.01 to 1%. Beyond their critical strains, the G′ values of the hydrogel tended to decrease with increasing strain % for both G’ and G”, indicating that the gels underwent deformation of the gel structure by gel–sol transition under the mechanical suppression and behaved as liquids at higher strain % when G’ = G”. With increasing pH, the G’ values at 0.01% strain tended to increase from G’ = 2.7 × 10^3^ Pa to 2.7 × 10^4^ Pa. At 2–3% strain, where G’ value began to decrease, the value of G” did to increase, indicating that the collapse of the gel structure reduced the stress relaxation into the gel. In the case of G’, the value increased tenfold when the pH was varied from 5 to 7, implying tight gelation appeared at higher pH, but as shown in Figure 4b, the tan δ = 1 appeared at lower strain % of 4% at pH 8, while the pH 7 occurred at 6% strain for the point of tan δ = 1. That is, it was a little stiffer at pH 8 than at pH 7, but the collapse of the gel structure occurred at a lower mechanical deformation in the case of pH 8. Figure 4b also showed that for the hydrogel adjusted at pH 5, the collapse of the gel structure could be maintained at about 12%, even when subjected to large mechanical deformation. However, the value of G’ was about 1/10 lower at 3 × 10^3^ Pa compared to 3 × 10^4^ Pa at pH 8. This indicated that the gel at pH 5 was softer and retained its gel structure against larger external deformation. When the pH was raised to 8, the G’ value increased somewhat. Additionally, as the Ca^2+^ concentration was changed, the values of G’ and G” were measured (Figure 4c,d) for 3% (*w*/*w*) pectin hydrogel adjusted to pH 7. It was noted that the G′ value improved significantly when the concentration of CaCl_2_ solution was increased from 0.9 M to 2.4 M. This might be caused by enhancing the crosslinking in higher Ca^2+^ concentrations. In the values obtained at 0.01% strain, varying 1 × 10^4^ Pa to 2.4 × 10^4^ Pa was observed from 0.9 M to 3.6 M, respectively. This was due to the high concentration of Ca^2+^ enhancing the stiffness of the hydrogel matrix. In the range of Ca^2+^ = 0.9 M to 2.4 M, the water content of the pectin hydrogel decreased from 1856 ± 15% to 1255 ±18%, respectively, and slightly increased to 1324 ± 16% at Ca^2+^ = 3.6 M. Moreover, the strain (%) at tan δ = 1 was 7% at 0.9 M and 2% for those gels at 3.6 M, respectively, suggesting that the hydrogels crosslinked at the high calcium concentrations were somewhat brittle. Furthermore, as seen in Figure 4e,f, the moduli change at various strains for hydrogels prepared from various pectin concentrations was raised from 1 wt% to 4 wt% in Ca^2+^ = 2.4 M at pH 7. The pectin concentrations of 1, 2, 3, and 4% had the G′ values at 0.01% strain, showing 4.3 × 10^3^ Pa, 9.9 × 10^3^ Pa, 2.6 × 10^4^ Pa, and 3.9 × 10^4^ Pa, respectively. When the concentration was 1 wt% the G’ modulus value showed a loose pectin network in the hydrogel, meaning a soft matrix. The hydrogel was stiffened due to the high polymer concentration gelling by crosslinking in the 4 wt% pectin. In the relationship between the strain (%) of hydrogels and tan δ = 1 as depicted in Figure 4e,f, the strain% at the gel point was changed from 8 % strain to 5% as the pectin component increased from 1% to 4%, respectively. Interestingly, the 3% pectin hydrogel exhibited the maximum strain of 9% at the gel transition point to liquid state, indicating that strong elasticity was maintained under the deformation.

Figure 5 shows frequency dependence on moduli for the pectin hydrogels. The resultant modulus was almost unchanged or slightly increased when the provided mechanical frequency increased in each hydrogel case. It was noted that the gel-like character was changed at pH 5 in the higher frequency of the provided mechanical strain, but mostly the used hydrogel had viscoelasticity independent of frequency over the accessible range of 0.01–100 (rad/s).

### 2.3. Characterization of Pectin Hydrogels for Changing Bulk pHs

It is known that pectin is characterized by the presence of D-galacturonic acid units in the polymer chain [27]. The crude pectin used in this study contained methyl ester groups at 74% and the remainder was hydrolyzed COOH groups of the COOCH_3_ ones. This chemical change is known to appear in the FTIR spectrum [28]. In the FTIR spectra of several pectin hydrogels, broad absorption bands at about 3300 cm^−1^ and 2941 cm^−1^ were ascribed to the stretching of OH and CH of CH_2_ groups in pectin methyl and methylene groups [29]. Due to the elimination of methoxy during the adjusting pH process, the comparison of pectins and the crude one by neutral water washing is made in Figure 6. This indicated that the band intensity at around 2941 cm^−1^ was reduced for all pectin hydrogel relative to the crude pectin. In the areas 1735–1750 and 1600–1640 cm^−1^ those were associated with the vibrations of the ester and carboxyl groups, respectively [28]. Especially in the basic pH region, a change in peak intensity was observed. This indicated that dissociation of the methyl ester group to the carboxylic acid group occurred readily in the basic pH conditions (Figure 6a). The intensity of peak at around 1745 cm^−1^ was higher than 1616 cm^−1^, indicating that a greater number of ester groups existed compared to carboxyl groups at pH 3.2. The peak absorption area dropped at 1745 cm^−1^ and rose at 1616 cm^−1^ for all pectin hydrogels at other pHs, suggesting that the dissociated carboxyl groups were generated under higher pH at 5, 6, and 7. In the case of pH 8, the significant improvement at 1616 cm^−1^ was seen compared to 1745 cm^−1^, indicating that almost ester groups underwent negatively charged COO^−^ in alkaline conditions [30]. The peaks at around 1402 cm^−1^ were attributed to the CH bending, whereas the peak at approximately 1255 cm^−1^ represented the C-O stretching of the O-C-O structure. The spectral region between 1200 and 800 cm^−1^ is considered the “fingerprint” region for each polysaccharide [31]. In addition, Figure 6b displays the FTIR spectra of pectin hydrogels in the presence of different amounts of crosslinker at pH 7. The results show no significant change in the intensity of these peaks at 1745 and 1616 cm^−1^ with a diversity of calcium concentrations in the range of 0.9 and 3.6 M. 

Because pH changes affected the de-esterification of pectin, the SEM observations of hydrogels were performed after lyophilization. The resultant view of the hydrogel surface and the cross-section of the matrix generated from pH 5 to 8 were measured at Ca^2+^ = 2.4 M. Figure 7a depicts SEM morphologies of the pectin hydrogel generated at different pHs. The obtained hydrogel at pH 7 looked like a denser morphology compared with that at pH 5, appearing sponge-like. This might be due to mostly COO^−^ groups being present at this pH. When the Ca^2+^ concentration was changed as shown in (b), the pictures emphasize dense crosslinking at higher calcium concentrations. Moreover, the distribution of pore voids became even and uniform in the size at pH 7, as Ca^2+^ = 2.4 M. When the calcium concentration was 3.6 M, less distribution of voids in the material surface was seen. This might be explained by the effect of the excess of calcium crosslinkers highly binding to the COO^−^ of pectin causing heterogeneity as well as limiting crosslinking. At lower pH than pH 7, owing to the limiting number of COO^−^ groups dissociated, the crosslinking process might follow point-crosslinking of ionic COO^−^ and Ca^2+^, and the hydrophobic COOCH_3_ groups might be influenced by the morphology more than the egg-box model because of less crosslinking between Ca^2+^ and the ionized carboxylic pectin groups [9,11]. 

## 3. Conclusions

By using viscoelastic measurements, the calcium pectin hydrogels influenced via various factors of pH, pectin contents, and Ca^2+^ concentrations were investigated. The pectin having 74% methyl estered pectin was extracted from pomelo albedo peel waste and used for calcium pectin gelation. The gelation time of the gel was examined by monitoring G’ and G” at different pHs and changing calcium ion concentrations. When the gelation time was defined as time at tan δ = 1, gelation occurred in about 7 min at pH 7 region but required more than 55 min at pH 3.2. Additionally, when the Ca^2+^ concentration was changed, gelation of pectin by calcium ions occurred more effectively in the neutral region than in the acidic region, resulting in a hydrogel with G’ of 7 × 10^2^ Pa compared to 2 × 10^2^ Pa at pH 5 after 55 min. Furthermore, as mechanical deformation was suppressed in the hydrogels, the hydrogel relaxed stress in the gel matrix up to about 1% strain, but over 1% strain, the gel structure deformed. G’ = 3 × 10^4^ Pa and G” = 4 × 10^3^ Pa were obtained at pH 7 and at 3% concentrations of pectin with 2.4 M CaCl_2_. When the pectin concentration was 1%, it was observed that G’ = 4 × 10^3^ Pa and G” = 6 × 10^2^ Pa, resulting in a softer hydrogel. The extent to which the gel structure breaks down under external mechanical deformation was found to occur at a greater deformation of 10% strain for the softer gels than for the stiffer gels.

## 4. Materials and Methods

### 4.1. Materials

Pomelo peels (flavedo and albedo) were collected from a supermarket in Bangkok, Thailand. Dextran standards (5000 to 670,000 g/mol) were purchased from Sigma. Calcium chloride was procured from Nacalai Tesque Inc., Kyoto, Japan. All other reagents used herein were of analytical grade.

### 4.2. Extraction of Pomelo Pectin

Firstly, the pomelo albedo peel was washed and dried in the oven at 60 degrees for 2 days to remove the water content inside, then ground to a powder before going to the extraction step. In this process, pectin was extracted using HCl 0.1 M [32] at 90 °C for 90 min and isolated by ethanol precipitation. Next, the crude pectin (extraction yield: 18%; DM: 74%) was freeze-dried for 24 h after washing by ethanol/water mixture (70/30 *v*/*v*) to separate the remaining acid residue.

Pectin solutions with a pH range from 5 to 8 were made by dissolving pectin powder in 2 mM phosphate-buffered saline (PBS) solution and shaking the mixture overnight to achieve a fully dissolved pectin solution. Pectin solution molecular weight was measured using GPC equipped with a refractive index (RI) detector (RID-10A; Shimadzu Corp., kyoto, Japan) according to the method by Huixin Jiang [33] with some modifications. The mobile phase (2 mM PBS pH 7.2) was eluted at a flow rate of 0.5 mL/min using a GPC column (KD-806 M; Shodex; Showa Denko K.K., Nagoya, Japan) at 50 °C. The molecular weights of the major fragment of pectin dissolved at pH 5, 6, 7, and 8 were 180 kDa, 176 kDa, 172 kDa, and 135 kDa, respectively.

Chemical structures of the hydrogels were investigated by FTIR spectroscopy using a JASCO FT/IR-4100 instrument (JASCO Corporation, Tokyo, Japan) in the 4000–500 cm^−1^ range with a resolution of 2 cm^−1^, and an average of 32 scans were acquired. The analysis was performed in triplicate.

### 4.3. Preparation of Pectin Calcium Hydrogel

In the first step, pectin powder was dispersed in DI water at room temperature for 24 h and adjusted pH using NaOH 1 M solution, then was centrifugated at 10 degrees Celsius and 6000 rpm for 15 min to obtain a visually clear solution. Secondly, the pectin precursor was slowly poured onto calcium solution and left overnight at room temperature to stabilize the gel networks as in Figure 3. 

### 4.4. Characterization of Calcium Pectinate Hydrogels

The influence of intrinsic factors on the viscoelastic behavior of calcium–pectinate hydrogels was investigated using a rheometer (Physica MCR 301, Anton Paar, Graz, Austria). The storage modulus (G’) and loss modulus (G”) were measured from 0.01% to 100% strain range at 1 Hz frequency at 25 °C. At each strain rate, the values of tan = G”/G’ were also recorded. The time dependence of storage and loss moduli was recognized by performing a time-sweep oscillation at 1 Hz and 1% strain for about 1 h. The measurements of time, amplitude, and frequency sweep enabled the study of gelation kinetics and gel mechanical characteristics [34,35]. To prevent slippage, a parallel-plate geometry (diameter 25 mm; gap 2.0 mm) was utilized in all rheological tests.

The water content of hydrogels [36,37] was determined by comparing the weight of the hydrogels before and after freeze-drying. The weight of wet samples was determined after carefully removing surface water. The hydrogels were then freeze-dried for 24 h to get dry weight. Parallel experiments were done 3 times. The water content of hydrogels was determined using the following formula: Water content = (W_0_ − W_1_)/ W_1_ × 100%,(1)
where W_0_ and W_1_ represent the weight of wet and dry hydrogels, respectively. 

The morphology of the hydrogel was investigated using scanning electron microscopy (SEM) JSM-5300 LV (JEOL, Tokyo, Japan). For the preparation of SEM samples, the pectinate hydrogels produced from 3 wt% pectin solutions with various calcium concentrations in different pHs were then frozen in the refrigerator for 24 h and freeze-dried for 24 h. After sputtering with gold using a rapid cool coater, the surfaces and cross-sections of the samples were inspected and photographed (Sanyu Denshi K.K., Nagoya, Japan).

## Figures and Tables

**Figure 1 gels-08-00814-f001:**
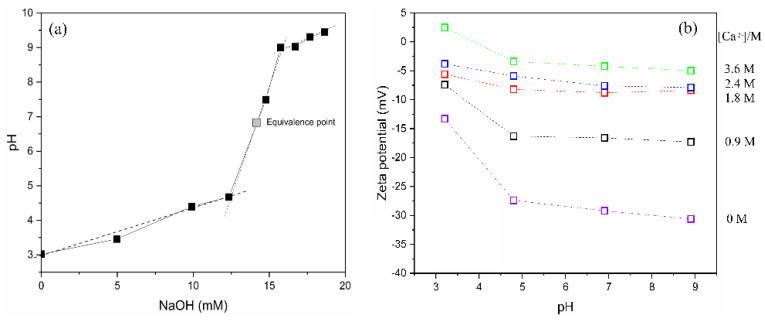
Titration (**a**) and zeta potential (**b**) experiments of pectin extracted from citrus pomelo waste.

**Figure 2 gels-08-00814-f002:**
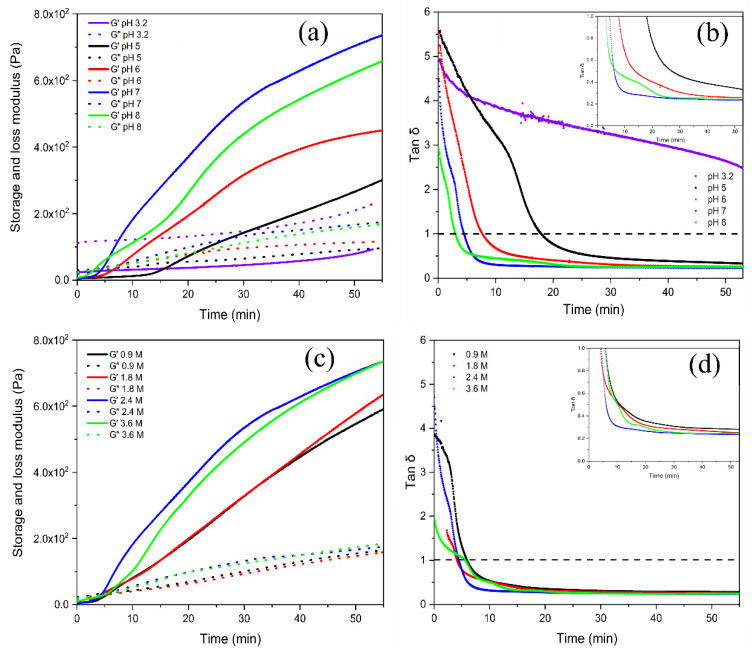
Time change of G’ and G” (**a**) and tan δ (**b**) for pectin (3% *w*/*w*) with CaCl_2_ = 2.4 M at various pHs (**c**,**d**) for 3% pectin solution at pH 7 with different calcium concentrations.

**Figure 3 gels-08-00814-f003:**
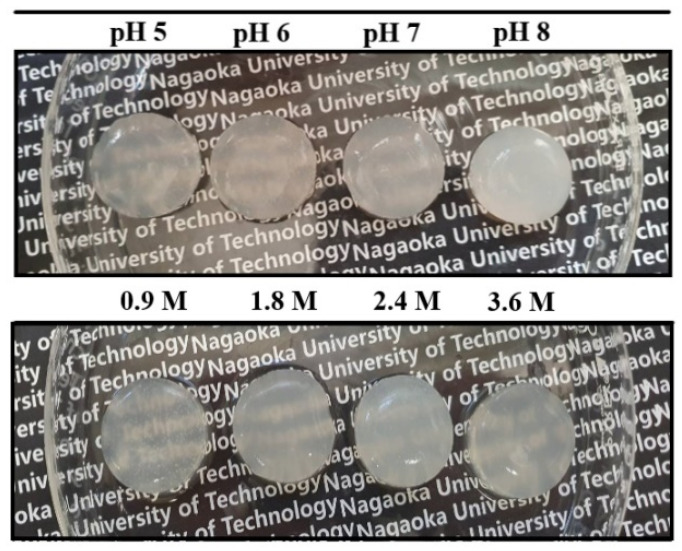
Appearance pictures of calcium–pectinate hydrogels prepared with different CaCl_2_ concentrations at various pHs.

**Figure 4 gels-08-00814-f004:**
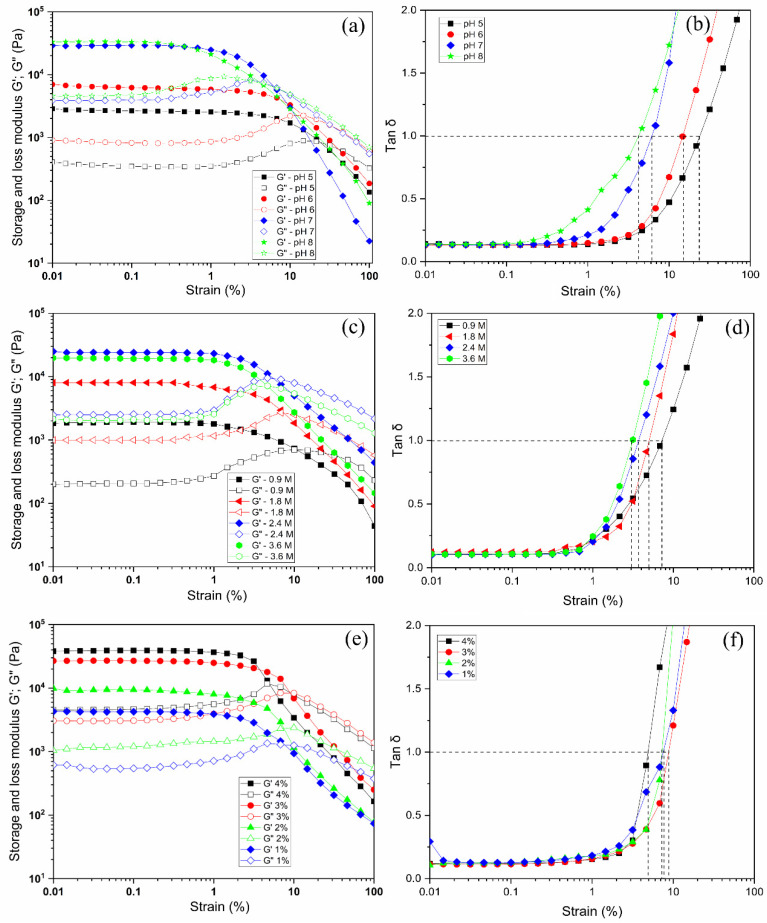
Strain sweep measurements of G’ and G” (left side) and tan δ (right side) for pectin hydrogels prepared at different pHs (**a**,**b**), various concentrations of calcium solution (**c**,**d**) for 3% pectin solution at pH 7, and pectin concentrations (**e**,**f**) for 2.4 M Ca^2+^ at pH 7.

**Figure 5 gels-08-00814-f005:**
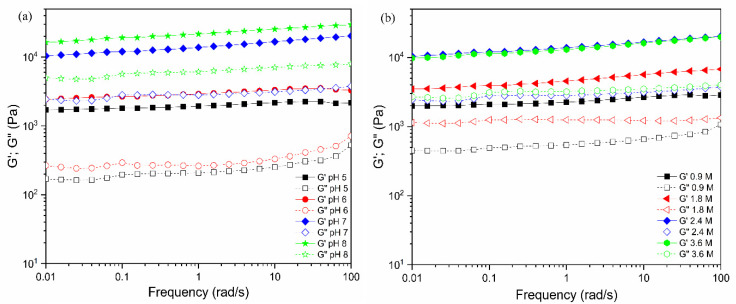
Frequency dependence of G′ and G″ moduli for pectin hydrogels prepared at different pHs (**a**) at 2.4 M Ca^2+^ and calcium concentrations (**b**) at pH 7. The pectin content used for the gelation was 3%. The moduli values were measured at strain 1%, 25 °C.

**Figure 6 gels-08-00814-f006:**
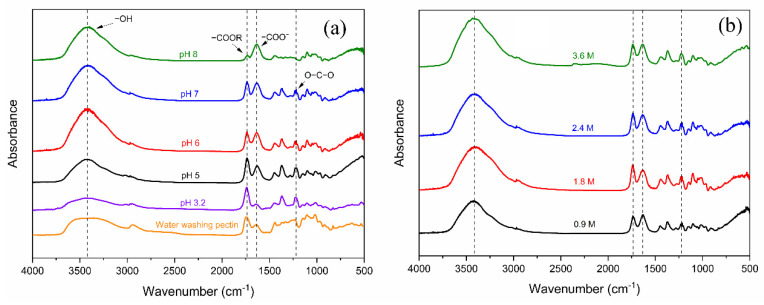
Chemical confirmation of pectin hydrogel at different (**a**) pH values and (**b**) calcium concentrations.

**Figure 7 gels-08-00814-f007:**
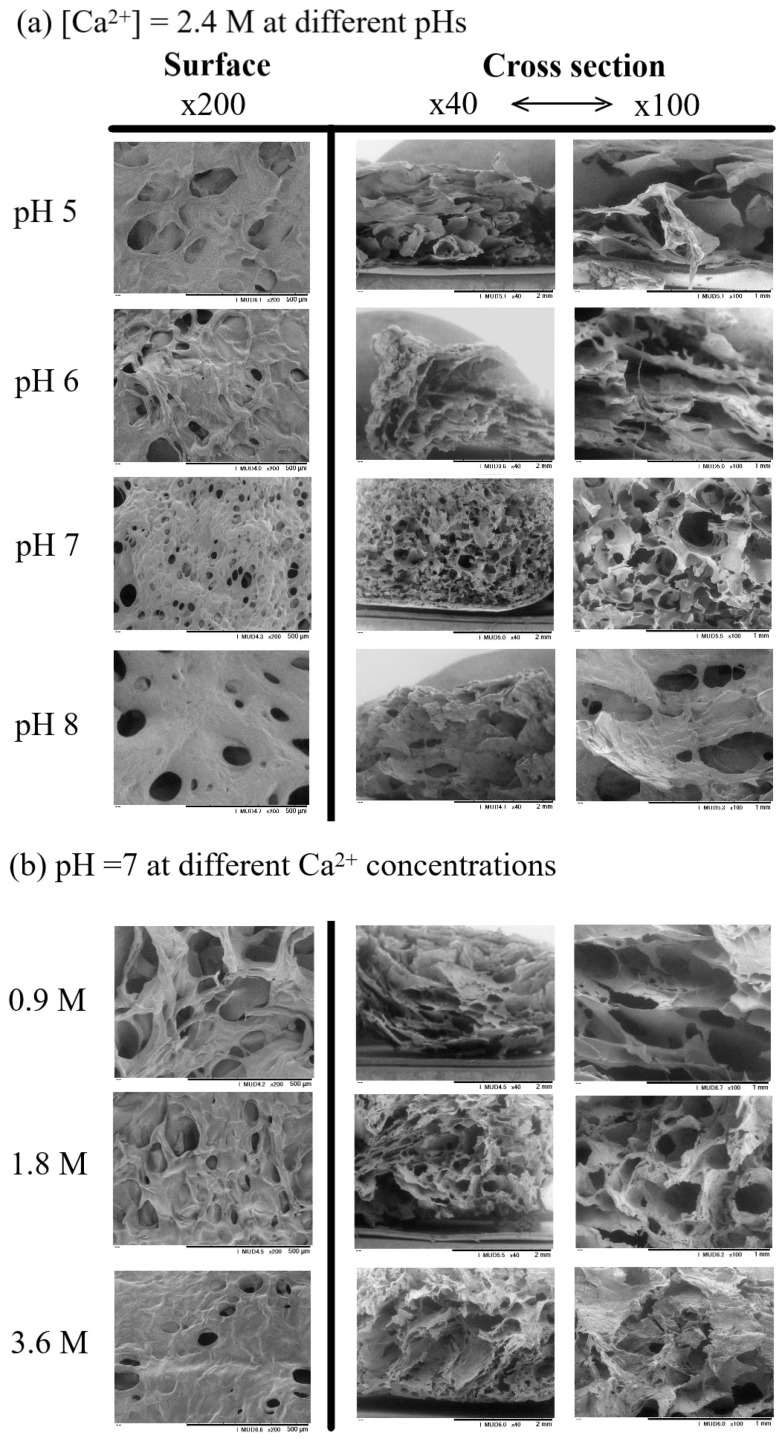
Surface and cross-section scanning electron microscopic (SEM) images of pectin hydrogel prepared at pH 5 to 8 (**a**); 0.9 M, 1.8 M, and 3.6 M Ca^2+^ solutions (**b**). Surface images in 200× *g* magnification. Cross-section images in 40× *g* and 100× *g* magnification.

**Table 1 gels-08-00814-t001:** Water content and gelation time of pectin hydrogel synthesized from various extrinsic parameters.

pH	Water Content (%) ^a^	Gelation Time (min) ^b^	Calcium Concentration (M)	Gelation Time (min) ^b^	Water Content (%) ^c^	% Pectin Solution	Water Content (%) ^d^
5	1557 ± 18	17.9	0.9	6.1	1856 ± 15	1	2757 ± 44
6	1329 ± 20	7.8	1.8	4.2	1632 ± 23	2	1904 ± 29
7	1255 ± 18	4.3	2.4	4.3	1255 ± 18	3	1255 ± 18
8	1207 ± 32	3.0	3.6	5.6	1324 ± 16	4	991 ± 16

^a^ Calcium concentration was 2.4 M and pectin content was 3%. ^b^ The gelation time was estimated at tan δ = 1. ^c^ Pectin content was 3% at pH 7. ^d^ Calcium concentration was 2.4 M at pH 7.

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
