# Peer review of "Viscoelastic Analysis of Pectin Hydrogels Regenerated from Citrus Pomelo Waste by Gelling Effects of Calcium Ion Crosslinking at Different pHs"

_gels, 2022, doi:10.3390/gels8120814_

Round 1

Reviewer 1 Report

Recommendation

Minor Revision

Comments:

This manuscript has reported the gelation behavior of hydrogels based on pectin which is extracted from citrus pomelo waste. The effect of calcium ions on the gelation under different pH conditions has been investigated using viscoelastic analysis. The gelation time and dynamic modulus properties of the hydrogels have been examined under mechanical strains ranging from 0.01 to 100 %. By monitoring the dynamic viscoelasticity values, the gelation occurs faster with increasing pH, leading to enhancement of the crosslink density. This research suggests that calcium ions cross-linkage with pectin affected the ionization of the carboxylic acid groups of pectin. The hydrogels constructed employing the extraction from food waste are interesting for investigation and the ample characterizations of the hydrogels are impressive. This manuscript, however, needs to be improved because of the vague conclusions. The authors should clarify what effects the pH, pectin contents, and Ca2+ concentrations have on the properties of the hydrogels in the sections of abstract, introduction and conclusion.

1) Line 71 on page 2: The effects of the pH and Ca2+ concentrations on the gelation process of the hydrogels need to be clarified in detail.

2) Line 141 on page 5: the water contents here seem to refer to swelling ratio, which can be over 100%. The water content is the mass fraction of water of the hydrogels, which should be less than 100%. Please check the definition of water content.

Author Response

Thank you for your suggestion. 

Reviewer 2 Report

This research is a repetition of very weak previous work that was published in other journals years ago. The idea of the research here has been addressed in tons of research over the past ten years, and the authors should have taken a new focus to search for improving the properties for example , 

Author Response

Thank you for your suggestion.

Reviewer 3 Report

1. In the introduction should be given about the previous related studies done

2. Still strengthen the significance and justification of the study 

3. What is pHs inline 71, write as various pH

4. Why the Minimum analysis of Oneway ANOVA can not do? and present the significant difference or similarity between the runs and present?

5. The text is presented continuously, and organized into paragraph wise based on the idea, then it will be attractive to the readers, 

6.  Better to enhance the discussion on SEM images try to discuss deep

7. Materials and methods should revise a little elaborative

8.  What is the source of the pectin extraction method used? 

9. Which variety of fruit peels are collected? 

10. For all the characterization mention the source of the methods

11.  Revisit the conclusions for fine-tuning

Author Response

Thank you very much for your suggestion.

Round 2

Reviewer 2 Report

Acceptable in the current form